# The development of adolescents' loneliness during the COVID-19 pandemic: The role of peer status and contact with friends

**Sofie J. Lorijn**[1]*, **Lydia Laninga-Wijnen**[2], **Maaike C. Engels**[3], **Gerine M. A. Lodder**[4], **René Veenstra**[1]

**1** Department of Sociology, University of Groningen, Groningen, The Netherlands, **2** Department of Developmental Psychology, INVEST Flagship, University of Turku, Turku, Finland, **3** Department of Educational Support and Innovation, Center for Information Technology, University of Groningen, Groningen, The Netherlands, **4** Department of Developmental Psychology, Tilburg University, Tilburg, The Netherlands

* s.j.lorijn@rug.nl

**Data Availability Statement:** The PRIMS data used in the paper will be made available for

## Abstract

The COVID-19 measures raised societal concerns about increases in adolescents' loneliness. This study examined trajectories of adolescents' loneliness during the pandemic, and whether trajectories varied across students with different types of peer status and contact with friends. We followed 512 Dutch students (Mage = 11.26, SD = 0.53; 53.1% girls) from before the pandemic (Jan/Feb 2020), over the first lockdown (March-May 2020, measured retrospectively), until the relaxation of measures (Oct/Nov 2020). Latent Growth Curve Analyses (LGCA) showed that average levels of loneliness declined. Multi-group LGCA showed that loneliness declined mostly for students with a victimized or rejected peer status, which suggests that students with a low peer status prior to the lockdown may have found temporary relief from negative peer experiences at school. Students who kept all-round contact with friends during the lockdown declined in loneliness, whereas students who had little contact or who did not (video) call friends did not.

## Introduction

Increased feelings of loneliness were a significant societal concern during the COVID-19 pandemic. Loneliness is a negative feeling that occurs when a person's perceived quantity or quality of social relationships is lower than desired [1]. Measures to prevent the spread of COVID-19, including physical isolation and school shutdowns, complicated social contact, which may have led to increased loneliness. Loneliness can lead to severe problems for health and functioning, including depression and low academic achievement [2, 3]. Expected increases in loneliness are thus alarming for individuals as well as societies. Particularly adolescents were expected to suffer from increased loneliness. The COVID-19 measures may have interfered with the maintenance of peer relationships of the desired quantity and quality, while peer relationships are of increased importance in adolescence [4, 5]. Recent studies have found an increase in adolescents' loneliness in response to the COVID-19 measures [6].

scientific research under conditions in 2024 here: https://doi.org/10.34894/U6XDT0.

**Funding:** This data collection is made possible by a grant awarded to René Veenstra, Herman van de Werfhorst, Jan Kornelis Dijkstra, Thijs Bol, and Sara Geven (2018) for the project 'Peer Relations in the Transition from Primary to Secondary school: Social, Behavioral and Academic Aspects of Social Integration' by the Netherlands Initiative for Education Research (NRO), grant number 40.5.18325.001. Visit the NRO website here: https://www.nro.nl/en. The funders had no role in study design, data collection and analysis, decision to publish, or preparation of the manuscript.

**Competing interests:** The authors have declared that no competing interests exist.

Despite producing valuable insights, prior work suffers from two shortcomings. First, many previous studies on adolescents' loneliness during the pandemic had several methodological limitations, given that they relied on cross-sectional designs [7], used small sample sizes [8], or assessed loneliness using a single-item measure [9]. Second, studies that applied more advanced designs predominantly examined *general changes* in loneliness, without identifying potential at-risk groups for loneliness during the pandemic. Heterogeneity in the findings of previous research, however, suggest individual differences in the effects of the COVID-19 pandemic on loneliness. Identifying which students were most at risk for loneliness during the pandemic is needed to develop targeted interventions for those who need it the most [10]. Although some researchers have focused on the impact of individual characteristics on heterogeneity of effects [11], peer-related characteristics have been neglected. Students' peer status prior to the pandemic, and the amount of contact they had with friends during the lockdown, may be key in predicting differences in loneliness. Students' peer status in school prior to the lockdown may predict to what extent not being around classmates during the school shutdown was socially harmful or beneficial, predicting changes in loneliness. Having contact with friends during the lockdown may have been a way for students to maintain their desired quantity and quality of social relationships, protecting them from feeling lonely.

This study built upon prior research by (1) longitudinally investigating trajectories of the construct of peer-related loneliness in a large sample of young adolescents during the COVID-19 pandemic, and (2) examining whether these trajectories differed between students depending on their pre-pandemic peer status and contact with friends during the lockdown.

## Loneliness during the COVID-19 pandemic

In the early stages of the COVID-19 pandemic, government agencies, the media, and health-care workers expressed their concerns about feelings of loneliness [12], in particular among young adolescents [13]. Drawing upon experiences during previous events leading to social isolation, for instance, in the context of the SARS virus, an increase in loneliness among adolescents was expected [14]. Whereas early studies on loneliness during the pandemic focused on adults [15–17], later studies increasingly focused on adolescents. A recent special issue on the impact of the pandemic on adolescents' adjustment confirms that many adolescents experienced higher levels of loneliness during the pandemic [6]. For instance, loneliness was found to have increased among (young) adolescents during the first lockdown in longitudinal studies in Belgium [9], Germany [11], Finland [18], and Israel [8]. This is consistent with a cross-sectional study among Canadian adolescents, who reported feeling more lonely during the first lockdown [7]. However, loneliness was found to remain low during the early period of the lockdown in Peru [19]. Moreover, the above-mentioned increases in loneliness were generally small and heterogeneous in their effects [6]. Thus, previous studies show some inconsistencies.

The COVID-19 pandemic may have been especially detrimental for adolescents transitioning from primary to secondary education. This transition brings along the challenge of entering a large, unfamiliar peer group. Previous work has shown that the transition poses a risk for adolescents' social lives [20], including increased social isolation [21] and increased loneliness [22]. It is likely that the negative impact of the pandemic on loneliness was exacerbated for students in this vulnerable period. Yet, previous studies have overlooked students in school transitions as an at-risk group during the pandemic. In order to further understanding of the impact of COVID-19 on loneliness, we examined loneliness trajectories over a period of almost a year, from before the start of the pandemic until a few months after students' transition from primary to secondary education. We expected that, on average, loneliness would increase from before

the pandemic (Jan/Feb 2020) over the first lockdown of the COVID-19 pandemic (March-May 2020), and the transition from primary to secondary education (Oct/Nov 2020).

## The role of peer status

Beyond the average increase in loneliness, the development of loneliness may differ per person, as indicated by the vast heterogeneity in loneliness found in previous studies [6]. Given that our focus was on peer-related loneliness, we expected that peer-related factors might predict heterogeneity in the development of loneliness over time. Two peer-related factors may matter in particular: students' pre-pandemic peer status and their contact with peers during the COVID period. These factors relate to students' fulfillment of their desired quantity and quality of social relationships, and thus their feelings of loneliness [1]. Students' peer status at school may give insight into the extent to which school shutdowns are likely to be harmful, beneficial, or have little impact on students' loneliness. Students' contact with friends during the lockdown may protect them from feeling lonely. Students with a lower peer status, referring to peer rejection [23] and victimization [24], are generally more lonely. In contrast, students with a higher peer status, referring to peer acceptance and popularity, experience fewer feelings of loneliness [23]. High- and low-status youth may have responded differently to the COVID-19 regulations, such as the school shutdown. For lower status youth (i.e., youth who scored relatively higher on rejection and/or victimization, and lower on popularity and/or acceptance) prior to the pandemic, three scenarios could occur. First, students with a lower peer status may have become less lonely during the COVID-19 lockdown. Decreased peer contact might have given them a 'break' from negative peer experiences, and having fewer or lower quality social interactions might have become more normative (i.e., other youth are 'in the same boat' now). Second, in contrast, students with a lower peer status may have experienced amplified loneliness during the pandemic. Despite having fewer negative peer experiences than when in school, these students may have poorer social skills and a lower quality social network, and so have difficulties maintaining their friendships outside of school; they might instead have used more avoidant coping strategies and withdrawn socially. In line with this, students who were more depressed and anxious before the pandemic received less support from friends during the pandemic [25]. In this scenario, risk factors for loneliness during the pandemic may be similar to risk factors for loneliness before the pandemic [26]. Third, students with a lower peer status may already have been less involved in their peers' lives *before* the pandemic. Therefore, the school shutdown would not have changed much for them, and their loneliness would have remained relatively stable over time. Taken together, we explored whether loneliness increased, decreased, or remained stable during the first lockdown for students with a lower peer status.

For students with a higher peer status prior to the pandemic, referring to youth who scored relatively lower on rejection and victimization, and higher on popularity and acceptance, two possible scenarios may arise. First, they may have remained stable in their low feelings of loneliness. Students with a higher peer status may be better equipped to maintain meaningful social contact as they can draw from their high-quality social network and well-developed social skills. Second, in contrast, loneliness may have increased for students with a higher peer status because they may have higher social needs, and they have more to lose socially as they enjoy positive social interactions. A greater desire for social contact combined with the restrictions on physical social contact, may have led to increased loneliness for high-status youth. In line with this reasoning, highly extraverted adolescents in particular were found to have experienced increases in loneliness during the pandemic [11]. In sum, two scenarios may have

occurred for high-status individuals: loneliness may either have increased or remained stable over the first lockdown.

## The role of contact with friends

In addition to peer status, contact with friends during the lockdown is a peer-related factor that may predict differences in trajectories of peer-related loneliness during the pandemic. Maintaining contact with friends may have protected against loneliness by helping students to obtain or maintain their desired quantity or quality of social relationships. As in-person contact was complicated by the COVID-19 lockdown, social contact was assumed to take place online. Indeed, some prior studies have shown that adolescents with fewer online connections, and less online contact with friends during the lockdown, experienced higher levels of loneliness [7, 8, 27, 28]. Most previous studies, however, did not include in-person contact; nor did they differentiate between types of online contact, referring to chat and (video) calls. Online social contact may be of lower quality than in-person contact [29], and therefore protect less against loneliness compared with in-person social contact. For instance, cue absence such as the lack of facial expression in text messaging or online chatting may complicate interpretation, leading to lower quality contact [29]. Thus, although online contact may have protected adolescents somewhat against loneliness during the pandemic, it remains unclear how the within-person constellation of different types of contact relate to loneliness. Based on the literature, we expected that loneliness would increase more for students with low (offline and online) contact with friends during the school shutdown.

## Control variables: Gender

Students' gender was included in the model to control for the possibility of gender differences in trajectories of loneliness. Even though gender is generally unrelated to loneliness [30], it is uncertain whether this was also true in the unique situation of the COVID-19 pandemic. For example, in a study among adults, women were more at risk for loneliness during the pandemic than men [26]. Moreover, we controlled for gender in the models that examined the influence of peer-related factors on loneliness, as boys are generally more rejected and victimized than girls [31].

## The present study

This study examined adolescents' trajectories of peer-related loneliness over the first lockdown of the COVID-19 pandemic (March-May '20), and the role of their peer status and social contact in these trajectories. First, we examined the average development of loneliness. We expected that, on average, loneliness would increase over the first lockdown (*Hypothesis 1*). Second, we explored the identification of peer status groups and contact profiles using a person-centered approach, which is desired when examining group differences in developmental trajectories [32]. Peer status groups were identified based on the extent to which students were *accepted*, *rejected*, *popular*, and *victimized* prior to the pandemic. The peer status groups give insight into students' general status in their classroom, as peer status is relatively stable over time [23]. Contact profiles were identified based on the amount of *offline contact*, *contact by (video) call*, and *contact by chat* during the school shutdown. Third, we examined how students in these varying peer status groups and contact profiles differed in their trajectories of loneliness. For students with *lower* peer status, we explored how loneliness developed during the first lockdown. For students with a *higher* peer status, we expected that loneliness would either increase (*Hypothesis 2a*) or remain stable (*Hypothesis 2b*) during the first lockdown. For the contact profiles, we expected that loneliness would increase more for students with low (offline

and online) contact with friends during the school shutdown, than for students with high (offline and online) contact with friends (*Hypothesis 3*). Our hypotheses, methods, and analysis plan were preregistered at the Open Science Framework (OSF) after the data collection but prior to the confirmatory analysis (see https://osf.io/c5je8).

## Methods

### Procedure and participants

Data stem from the first cohort of the PRIMS project (an acronym for transition from PRImary to Secondary school). PRIMS initially aimed to investigate the role of peers in the transition from primary to secondary education; its focus shifted to the role of peers in the COVID-19 pandemic after the onset of the pandemic. Students were followed from their last year of primary education to their first year of secondary education. The first survey was conducted in January and February 2020, before the onset of the pandemic in the Netherlands, when the students were in their final year of primary education. This serves as a baseline measure for loneliness. The second survey was conducted in October-November 2020, roughly six weeks after the students transitioned to secondary education, and included questions about two time points. First, this survey included retrospective questions on the period between March and May 2020, during the first lockdown in the Netherlands, when the students were in their last year of primary education. We asked how the students remembered having experienced the lockdown. Second, this survey included questions on October-November 2020, when the students filled out the second survey. Thus, in total, the data cover three time points: a pre-pandemic assessment, a retrospective assessment of loneliness and contact *during* the pandemic, and a post-first-lockdown assessment.

The COVID-19 pandemic hit the Netherlands in February 2020. The first lockdown began on 16 March 2020. The population was advised to stay at home and keep physical distance. For adolescents, the lockdown entailed being home-schooled and not attending hobbies, such as sports. Our sample was in the final year of primary education. For them, the lockdown also meant the cancellation of the standardized exit test, farewell activities at their primary school, and introductory activities at their new secondary school. This lockdown was followed by relaxations, with schools reopening partially from 11 May and opening fully in early June. In October-November 2020, most measures were lifted and students had returned to school. Yet, there was still a 'partial lockdown', with measures such as being able to receive a maximum of three visitors at home, cancellation of events, and physical distancing still in place.

We selected schools based on a stratified sample design, with the sample frame consisting of the full population of Dutch primary schools. Based on the expected response rate, 339 schools were sampled and approached for participation. Sixty-six of these schools (19.5%), totaling 105 classes, were willing to participate after they received information about the study, resulting in a representative sample on region, level of urbanization, socio-economic composition, and test scores at the school level. We invited all students in the Dutch grade 8 (11–12 years) of the participating schools to take part in the study. Of the 2,489 students invited, 1,535 (62.1%) received active informed written parental consent to participate. A total of 1,474 students filled out the first survey (96.0% of the students with consent). The remaining 61 students did not complete the survey because they were absent from school or experienced technical problems with the survey. The students filled out the online questionnaire, of approximately 45 minutes, during regular school hours under the supervision of their teacher. For more information on sampling and data collection, see [33].

Of the 1,474 students who participated in the first survey, 892 (60.5%) provided valid contact details and were approached to participate in the second survey. In total, 583 (65.4%)

students filled out the second survey; this is 39.6% of the 1,474 participants of the first survey. This survey was shortened to minimize students' efforts during the pandemic. Students filled out this survey at home; it took approximately 15 minutes. As an incentive, students who completed the survey could join a lottery to win a Nintendo Switch. The PRIMS project received ethical approval from the University of Amsterdam (project number 2019-AISSR -10381).

## Missing data

There were two types of missing data in this study. First, there was missing data because of attrition, reducing the sample from the 1,474 students who participated in the first survey to the 583 students who participated in both surveys. Second, there was missing data by design. To obtain a reliable picture of students' peer status, classrooms with a lower participation rate than 40% were excluded from the analysis [34], further reducing the sample from 583 to 512 students. The non-respondents ($n = 1,474–512 = 962$) differed from the respondents ($n = 512$) in having lower socioeconomic backgrounds $t(1323) = -2.550, p = .011; d = 0.15$, and being less accepted by peers $t(1133.32) = -2.526, p = .012; d = 0.14$. No significant differences were found for gender, popularity, rejection, victimization, or loneliness. In sum, our subsample consisted of 512 students from 29 classes and 18 schools, including 34.7% of the original PRIMS respondents from the first survey. Classrooms varied, having from 9 to 33 participating students ($M = 24.96, SD = 3.97$) and from 40% to 100% participation rate ($M = 72.71, SD = 17.03$). In the main analysis, partially missing values were handled using the full information maximum likelihood estimate, which is an unbiased method for handling missing data that is not at random [35].

## Measures

**Loneliness (all time points).**   We measured loneliness using the subscale of peer-related loneliness of the Loneliness and Aloneness scale for Children and Adolescents (LACA) [36]. We selected five of the original 12 items based on highest factor loadings of previous studies [37–39]. The items were, 'I feel excluded by my classmates', 'I feel alone at school', 'I feel abandoned by my friends', 'I feel left out by my friends', and 'I feel sad because I have no friends'. Students rated the items from 0 = *never* to 3 = *always*. The scale was generated by averaging the answers to the items for every participant who gave a valid response to at least two items. The scale revealed a good reliability score over the three time points (all $\alpha \geq .84$). Loneliness was assessed at all three time points, with the measure in March-May '20 being measured retrospectively. We adapted the items of the retrospective scale slightly to suit the circumstances of the school shutdown. Specifically, we changed the item, 'I feel alone at school' to 'I felt alone'. The other items were rephrased to the past tense. The retrospective items were presented after the following introductory statement: '*The following questions are about the school shutdown because of the corona virus. Think back to the time you could not go to school because of the corona virus.*'

As the meaning of this scale needed to be the same across repeated measures in order to optimally measure trajectories of loneliness, measurement invariance was assessed [40]. We included configural, metric, and scalar steps in our test of measurement invariance. The configural step showed acceptable fit for three of the fit indices (CFI = .89, TLI = .87, RMSEA = .10, SRMR = .06). The fit for the metric step was questionable, with two of the indices showing inadequate fit (CFI = .88, TLI = .87, RMSEA = .10, SRMR = .07). The scalar step showed unacceptable fit (CFI = .85, TLI = .85, RMSEA = .11, SRMR = .08). This means that the loneliness measure did not have the exact same meaning at each time point, which may be due to the retrospective measure during the school shutdown and the slight changes to items on the scale

for this measure. Moreover, the items contributing to feelings of loneliness may load onto loneliness differently for the pandemic. For instance, feeling alone may have become more prevalent during the pandemic, but may not have contributed to loneliness as much as before the pandemic, because feeling alone during a lockdown is a result of external factors rather than social rejection. Despite not finding measurement invariance for the scalar step, we include the retrospective measure because this is the best available measure of loneliness for the lockdown. Moreover, although measured in the same survey, feelings of loneliness were only moderately related between March-May '20 and Oct/Nov '20 ($r = .51$). As a result, we should interpret the outcomes for the school shutdown with more caution; this is discussed in the limitations section.

**Peer acceptance, peer rejection, and popularity (Jan/Feb '20).** Peer acceptance, rejection, and popularity were assessed using unlimited classroom-based peer nominations. Students were asked, 'Which classmates do you like?', 'Which classmates do you dislike?', and 'Which classmates are popular?'. A list of all classmates was provided. Children in the Netherlands have one classroom in their final year of primary education and do not switch groups. The number of nominations received was divided by the number of possible nominations, being the total number of classmates filling out the survey minus 1. This means that nomination scores account for differences in class size and participation rates. This yielded proportion scores for peer acceptance, rejection, and popularity, ranging from 0 to 1, with more nominations resulting in higher scores.

**Victimization (Jan/Feb '20).** Victimization was measured by asking, 'Can you indicate how often you have been bullied at school in the past few months?' Before answering this question, students watched an introduction clip providing a definition of bullying [41, 42]. Students rated the question from 1 = *never* to 5 = *several times a week*. To ease interpretation for comparing peer status predictors, we recoded the scale into 1 = 0; 2 = .25; 3 = .50; 4 = .75; and 5 = 1.

**Contact with friends (March-May '20, retrospective measure).** Contact with friends during the school shutdown was assessed using three questions. Students indicated how often they (1) saw friends face-to-face during the school shutdown, (2) (video) called friends during the school shutdown, and (3) chatted online with friends during the school shutdown. Online chatting included text messaging, such as through WhatsApp or Snapchat. Students rated the items from 0 = *daily* to 4 = *never;* answer categories were reverse-coded so that scores ranged from *never* = 0 to *daily* = 4.

**Gender (Jan/Feb '20).** Students indicated whether they were a boy, girl, or other. Because only two students (0.1%) indicated being 'other', we coded 'other' as missing, and gender as 0 = *boy*; 1 = *girl*.

## Analytic strategy

We performed statistical analyses to test the hypothesis in M*plus* version 8.5, in three steps [35]. First, to test Hypothesis 1, we analyzed the general linear development of loneliness using Latent Growth Curve Analysis (LGCA) in two models. In the first model, the trajectory was estimated using an unconditional (i.e., containing no predictors) LGCA. In the second model, gender was included to test if it significantly added to the model and thus should be controlled for. We used multiple indices to assess model fit: the Comparative Fit Index (CFI), Tucker-Lewis Index (TLI), the Root Means Square Error of Approximation (RMSEA), the standardized root mean squared residual (SRMR), and $\chi^2$ [43]. For CFI and TLI, $\geq$.90 is considered acceptable and $\geq$.95 reflects a good model fit. The RMSEA should not exceed .08, and shows a good fit for values of $\leq$.06. An SMRS of $\leq$.06 is considered acceptable, and $\leq$.05 reflects a good fit. The loading of the slope factor for loneliness in Jan/Feb '20 (T1) was fixed to 0, so that

the mean intercept reflected the level of loneliness in Jan/Feb '20. For the measures in March-May '20 and Oct-Nov '20, the loadings of the slope factors were fixed to 1 and 2, respectively.

Second, we explored peer status groups and contact profiles using Latent Class Analysis (LCA). Peer status groups were identified based on the within-person constellation of peer acceptance, rejection, popularity, and victimization in Jan/Feb '20 (T1). Contact profiles were identified based on the within-person constellation of frequency of contact with friends in-person, by (video) calling, and by chatting during the school shutdown in March-May '20 (T2). The optimal number of latent classes was determined using a combination of criteria. The sample-size adjusted Bayesian Information Criterion (aBIC) shows whether adding a class improves the model fit, with the lowest aBIC value indicating the best fit. We used the sample-size adjusted BIC, because BIC may underestimate the preferred number of classes when relying on small sample sizes. Entropy ($E$) is an omnibus index ranging from 0 to 1, with a value of $\geq .80$ indicating a good fit. The Lo-Mendell-Rubin Likelihood Ratio Test (LMR-LRT) indicates if there is a significant improvement in model fit when a class is added.

Third, we examined whether variations in trajectories of loneliness were predicted by the peer status groups and contact profiles, respectively, using two multi-group LGCAs (one for peer status groups and one for contact profiles). For both LGCAs, we compared a constrained multi-group model with an unconstrained multi-group model, by testing significance in $\Delta \chi^2$ and $\Delta$CFI $< .005$. The constrained model held intercepts and slopes equal across groups, assuming that all groups were similar in loneliness. In the unconstrained model, intercepts and slopes were estimated freely across groups, allowing groups to vary in loneliness. If the unconstrained model were to fit better, this would indicate that groups differed in their intercepts and slopes. In this case, all possible combinations of constraints were tested to identify where the groups differed and to determine the final, best-fitting model. Model constraints were justified if the unconstrained model was not significantly better than the partly constrained model in $\Delta \chi^2$ and if the $\Delta$CFI was $< .005$. Model fit of the final model was determined using $\chi^2$, RMSEA, SRMR, and CFI. Overlap in 95% confidence intervals of the intercepts and slopes between groups were examined to test whether groups differed significantly from each other. If the model in the first step (the general LGCA) indicated that gender should be added to the model as a covariate, it was important to take into account that the retrieved intercept and slope values for the multi-group models were conditional on the reference group (in our case, the reference group was boys). We ran the same model ($n = 512$) with girls as the reference group, to calculate the intercepts, slopes, and confidence intervals for girls. Results are shown for both boys and girls as the reference group, relying on the same model to ease interpretation for the reader in determining intercepts and slopes per gender. In all analyses, we controlled for clustering of students at the classroom level (i.e., students are nested in classrooms), using the measure in Jan/Feb '20 with the "complex analysis" option and "cluster" command in M*plus*. The maximum likelihood estimator (MLR) was applied to handle potential non-normal distributions [44].

## Results

### Descriptive statistics

Table 1 presents the descriptive statistics and correlations of the study variables. On average, loneliness was low and declined from 0.27 before the COVID-19 pandemic to 0.21 during the first lockdown of the COVID-19 pandemic, and 0.12 after the reopening of schools. Students' feelings of loneliness pre-pandemic moderately related to their loneliness during the pandemic ($r = .55$), and loneliness during the pandemic moderately related to loneliness after the reopening of schools ($r = .51$). Furthermore, all types of peer relationships were

**Table 1. Descriptive statistics and correlations for study variables.**

| Variables | N | 1 | 2 | 3 | 4 | 5 | 6 | 7 | 8 | 9 | 10 | Mean (SD) |
|---|---|---|---|---|---|---|---|---|---|---|---|---|
| **Peer status** | | | | | | | | | | | | |
| 1. Acceptance | 512 | - | | | | | | | | | | .49 (.18) |
| 2. Rejection | 512 | -.55*** | - | | | | | | | | | .11 (.14) |
| 3. Popularity | 512 | .21*** | -.10* | - | | | | | | | | .12 (.16) |
| 4. Victimization | 508 | -.26*** | .31*** | -.12** | - | | | | | | | .07 (.19) |
| **Loneliness** | | | | | | | | | | | | |
| 5. Pre-lockdown | 508 | -.28*** | .30*** | -.18*** | .47*** | - | | | | | | 0.27 (0.44) |
| 6. During the lockdown | 505 | -.27*** | .30*** | -.14** | .29*** | .55*** | - | | | | | 0.21 (0.42) |
| 7. Post-lockdown | 496 | -.09* | .15*** | -.13** | .24*** | .42*** | .51*** | - | | | | 0.12 (0.30) |
| **Peer contact** | | | | | | | | | | | | |
| 8. Contact in person | 505 | .06 | -.08 | .12** | -.08 | -.16*** | -.17*** | -.06 | - | | | 2.27 (1.16) |
| 9. Contact (video) call | 505 | .01 | -.00 | .11* | -.02 | -.08 | -.06 | -.05 | .07 | - | | 2.43 (1.34) |
| 10. Contact chat | 505 | .02 | -.06 | .13** | -.03 | -.09* | -.17*** | -.12** | .24*** | .43*** | - | 3.52 (0.76) |
| 11. Gender | 511 | .08 | -.06 | -.12** | -.03 | .15*** | .14*** | .17*** | -.07 | .20*** | .08 | 53.2% girls |

*Note.* *$p \leq .05$

**$p \leq .01$

***$p \leq .001$. Peer status (1–4) measured in January/February 2020. Peer contact (8–10) measured retrospectively in October/November 2020 for the lockdown period (March-May 2020).

related to loneliness at all time points. Acceptance and popularity were weakly related to lower levels of loneliness, whereas rejection and victimization were weakly to moderately related to higher levels of loneliness. Associations between acceptance, rejection, popularity, and victimization showed that these peer experiences were weakly to strongly related to each other, justifying a person-centered approach.

## Average trajectory of loneliness

First, we specified an unconditional latent growth model for loneliness, which revealed excellent model fit, $\chi^2(1, N = 512) = 1.37$, $p = .242$, RMSEA = .027, CFI = .997, SRMR = .012. The significant negative slope ($M_S$ = -0.08, [-0.10; -0.06], $p < .001$) indicated that students' loneliness on average decreased during the first lockdown (March-May 2020), contrary Hypothesis 1. Exploring separate items of the loneliness scale provided more insight into the decrease of loneliness during school shutdown (March-May '20), with students feeling more alone and sad because they had no friends, but less excluded, abandoned, and left out. There was a strong negative correlation between the intercept and slope ($r$ = -.72, $p < .001$), denoting that students with higher initial levels of loneliness tended to have a more rapid decrease in loneliness over time. Significant variances around the intercept and slope ($D_i$ = 0.15, $p < .001$; $D_s$ = 0.03, $p = .004$) indicate variation in the initial level and trajectory of loneliness.

Second, we added gender to the LGCA to determine if gender should be included as a covariate. When gender was added, all model fit indices improved, $\chi^2(2, N = 512) = 1.31$, $p = .521$, RMSEA < .001, CFI > .999, SRMR = .009. The model fit thus remained excellent. The unconditional model did not have a better fit than the model including gender ($\Delta\chi^2(1) = 0.02$, $p = .877$). Moreover, even though gender did not predict differences in the average trajectory of loneliness ($p = .357$), the average initial level of loneliness was significantly higher for girls ($p = .001$). Gender thus added to the model and was included as a covariate in the following models.

## Peer status groups

Latent class analysis clustered students in peer status groups based on acceptance, rejection, popularity, and victimization. The three-group solution was found to be the optimal model based on aBIC, $E$, and LMR-LRT fit indices, as explained in S1 Appendix. Three groups were identified: a (1) *normative*, (2) *rejected*, and (3) *victimized* peer status group. Fig 1 illustrates the means per peer status group. Students in the *normative* group (84.0%) had higher levels of acceptance and popularity, and lower levels of rejection and victimization compared with students in the other groups. We called this group the *normative* group because it was the largest group and its peer status dimensions were closest to the average. Students in the *rejected* group (11.3%) had relatively low levels of acceptance, average levels of popularity and victimization, and particularly high levels of rejection. Students in the *victimized* group (4.7%) had relatively low levels of popularity and acceptance, high levels of rejection, and particularly higher levels of victimization compared with the other groups.

## Trajectories of loneliness per peer status group

Multi-group LGCA was applied to examine differences in trajectories of loneliness per peer status group. Based on $\Delta\chi^2$ and $\Delta$CFI fit indices, the unconstrained model was used as the final model, as explained in S1 Appendix. Table 2 presents the parameter estimates of the final model. Students in the *victimized* peer status group had the highest initial levels of loneliness, followed by the *rejected* group and the *normative* group. All peer status groups showed a significant decrease in loneliness over time. The findings for boys and girls relied on the same model, for which we alternated the reference group to ease interpretation for the reader in determining intercepts and slopes per gender. Focusing on initial levels of loneliness, boys in the *victimized* group ($M_I = 0.83$, [0.41; 1.26]) had higher initial levels of loneliness than boys in the *normative* group ($M_I = 0.15$, [0.11; 0.20]) and the *rejected* group ($M_I = 0.29$ [0.17; 0.40]). We found no difference between boys in the *rejected* group and boys in the *normative* group. The initial levels of loneliness for girls in the *rejected* group ($M_I = 0.64$, [0.46; 0.82]) and the *victimized* group ($M_I = 0.81$, [0.43; 1.19], $p < .001$) did not differ from each other, and these levels were higher than the initial levels of loneliness in the *normative* group ($M_I = 0.26$, [0.19; 0.33]).

Students with a lower peer status had steeper declines in loneliness during the first lockdown than students with a normative peer status. Specifically, boys in the *victimized* group ($M_S = -0.27$, [-0.36; -0.19]) had steeper declines in loneliness than boys in the *rejected* ($M_S = -0.09$, [-0.15; -0.04]) and the *normative* ($M_S = -0.05$, [-0.07; -0.03]) groups. We found no difference in the decline in loneliness between the *rejected* ($M_S = -0.19$, [-0.29; -0.08]) and the *victimized* groups ($M_S = -0.36$, [-0.515; -0.206]) for girls. The decline for girls in the *rejected* and the *victimized* groups was larger than the decline for girls in the *normative* group ($M_S = -0.06$, [-0.09; -0.03]).

For students with a *higher* peer status, we expected that loneliness would either increase (*Hypothesis 2a*) or remain stable (*Hypothesis 2b*). Although the LCA did not detect a specific very high peer status group (i.e., a popular-liked group), we tested this hypothesis using the *normative* group because this group had higher levels of acceptance and popularity, and lower levels of rejection and victimization than the other groups. The significant negative slope for boys ($M_S = -0.05$, [-0.07; -0.03]) and girls ($M_S = -0.06$, [-0.09; -0.03]) in the normative group shows that loneliness decreased for students with a higher peer status, contrary to Hypothesis 2a and 2b. Fig 2 visualizes the relative decline in loneliness for each peer status group, for boys and girls separately. It shows that the *victimized* group had the highest levels of loneliness at all time points, but also the steepest decline in loneliness over time.

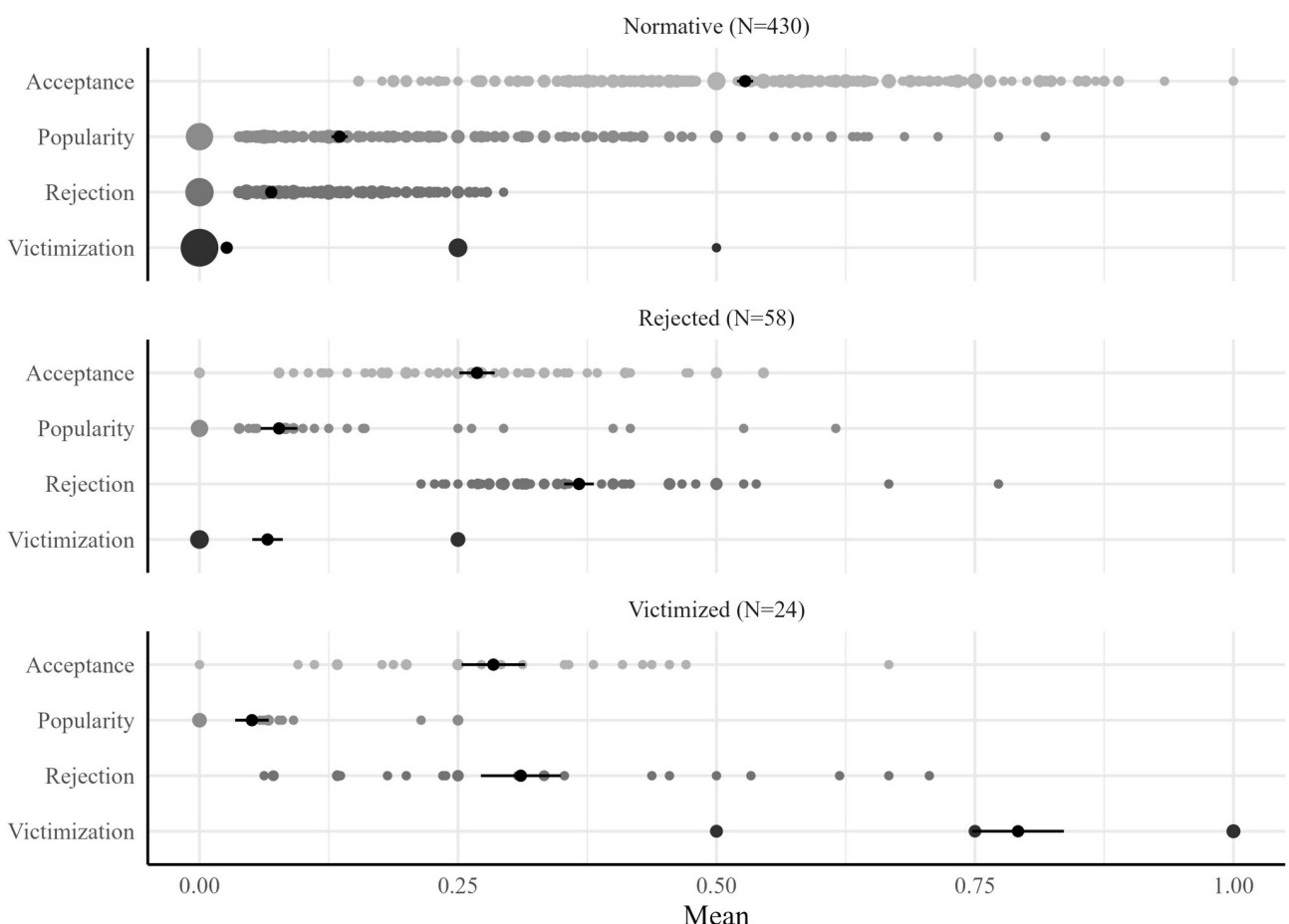

**Fig 1. Means and 95% CIs of acceptance, popularity, rejection, and victimization per peer status group.** *Note.* The dots show overlapping data points for small (*n* = 100), medium (*n* = 200), and large (*n* = 300) numbers of observations per data point.

To better understand the role of peer status groups in declines of loneliness, we conducted additional analyses to test absolute differences in levels of loneliness at each time point, using (multivariate) analysis of covariance. These analyses revealed that all peer status groups differed in (combined time points of) loneliness when gender was included ($F$ (6, 970) = 16.32, $p$ < .001, Wilks' = 0.825). Analysis of covariance showed that this difference existed at all time points, in Jan/Feb '20: $F$ (2, 503) = 46.80, $p$ < .001, $\eta p^2$ = .18, in March-May '20 (measured retrospectively): $F$(2, 500) = 24.95, $p$ < .001, $\eta p^2$ = .09, and in Oct/Nov '20: $F$(2, 491) = 8.24, $p$ <

**Table 2. Intercepts and slopes of the multi-group LGCA on loneliness per peer status group by gender (N = 512).**

| Gender | Parameter | | Peer Status Group | |
|---|---|---|---|---|
| | | Normative | Rejected | Victimized |
| Boys | *M* intercept | 0.154 [0.114; 0.195] *** | 0.285 [0.169; 0.401] *** | 0.832 [0.407; 1.256] *** |
| | *M* slope | -0.051 [-0.073; -0.028] *** | -0.094 [-0.151; -0.037] *** | -0.273 [-0.361; -0.185] *** |
| Girls | *M* intercept | 0.261 [0.192; 0.329] *** | 0.637 [0.455; 0.819] *** | 0.805 [0.426; 1.185] *** |
| | *M* slope | -0.060 [-0.092; -0.029] *** | -0.186 [-0.291; -0.082] *** | -0.361 [-0.515; -0.206] *** |

*Note.* *p* < .05

***p ≤ .001. Numbers in parentheses represent 95% confidence intervals.

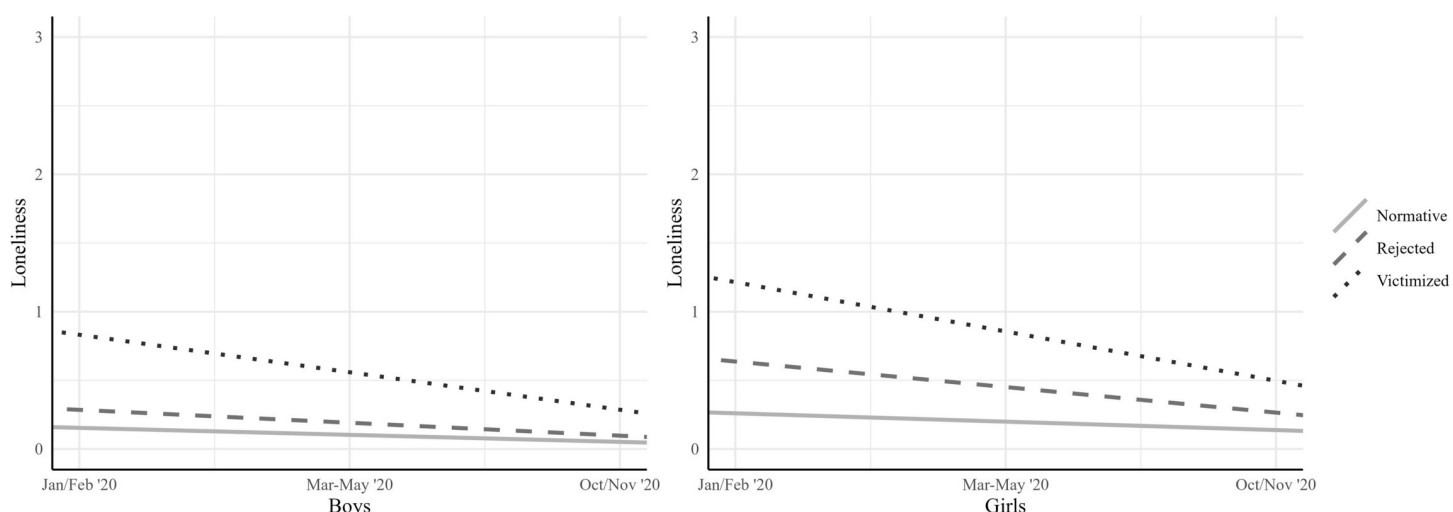

**Fig 2. Trajectories of loneliness per peer status group per gender.** *Note.* Loneliness during the first lockdown in March-May 2020 was measured retrospectively in October/November 2020.

.001, $\eta p^2$ = .03. Post-hoc tests using a Bonferroni adjustment revealed that in Jan/Feb '20, all groups differed in loneliness (all $p < .001$), with the *victimized* group reporting the highest levels of loneliness ($M = 0.96$), followed by the *rejected* group ($M = 0.44$), and the *normative* group ($M = 0.21$). In March-May '20, students in the *victimized* ($M = 0.57$) and the *rejected* groups ($M = 0.43$) did not differ from each other in loneliness, but both groups reported higher levels of loneliness than the *normative* group ($M = 0.16$, both $p < .001$). A similar pattern was visible in Oct/Nov '20, but the difference in loneliness between students in the *victimized* group ($M = 0.31$) and the *normative* group ($M = 0.10$, $p = .001$) was significant.

## Contact with friends

To explore how contact with friends during the lockdown related to trajectories of loneliness, we examined the type (i.e., offline or online) and amount of contact with friends during the school shutdown. Table 3 shows that, despite the physical isolation measures, most students saw their friends in person weekly to daily (52.7%). Around one-fifth of the students (23.6%) saw their friends less than once a month or did not meet their friends in person over the school shutdown. Most students had online contact with friends during the school shutdown, especially by chat (including text messaging).

Next, we conducted LCA to explore contact profiles. The three-group solution was found to be the optimal model based on aBIC, E, and LMR-LRT fit indices, as explained in S2 Appendix. Students were clustered into: (1) *low* contact, (2) *high contact but low (video) calling*, and (3) *high all-round* contact profiles. Table 4 shows that students in the *low* contact profile (6.1%) had low offline and online contact with friends during the lockdown. Students in the *high contact but low (video) calling* profile (19.8%) saw their friends monthly to weekly, and

**Table 3. Percentages of type and amount of contact with friends.**

|  | Never | Less than once a month | Monthly | Weekly | Daily |
|---|---|---|---|---|---|
| Contact in person | 12.1% | 11.5% | 23.8% | 43.2% | 9.5% |
| Contact by (video) call | 16.0% | 8.1% | 13.1% | 42.0% | 20.8% |
| Contact by chat | 0.8% | 2.8% | 3.0% | 30.9% | 62.6% |

**Table 4. Means and 95% CIs of contact in person, contact by (video) call, and contact by chat per contact profile.**

| Contact profile | Means | | |
|---|---|---|---|
| | Contact in person | Contact by (video) call | Contact by chat |
| Low contact (6.1%) | 1.30 [0.95; 1.65] | 0.98 [0.54; 1.42] | 1.25 [0.95; 1.55] |
| High contact but low (video) calling (19.8%) | 2.29 [2.04; 2.55] | 0.42 [0.28; 0.56] | 3.37 [3.26; 3.47] |
| High all-round contact (74.1%) | 2.33 [2.20; 2.46] | 3.12 [3.05; 3.20] | 3.74 [3.69; 3.79] |

had weekly to daily contact by chat, yet they hardly (video) called friends. Students in the *high all-round* contact profile (74.1%) saw their friends in person monthly to weekly over the school shutdown and had weekly contact with friends by (video) call and chat during the school shutdown. Table 5 shows the overlap between the contact profiles and the peer status groups. Although students in the *rejected* peer status group seem to be overrepresented in the *low* contact profile, the contact profiles did not differ significantly in their representation of the peer status groups, as shown by the Fisher-Freeman-Halton Exact test (6.81, $p = .127$).

## Trajectories of loneliness by contact profile

Multi-group latent growth curve models were used to determine if the contact profiles differed in trajectories of loneliness over the first lockdown (March-May 2020). The unconstrained model in which intercepts and slopes for contact profiles were freely estimated was used as the final model based on $\Delta\chi^2$ and $\Delta$CFI fit indices, as explained in S2 Appendix. Again, we ran the same model twice, with boys and girls as reference group, respectively, to ease interpretation of effects conditional on gender (Table 6).

The slopes in these models were evaluated to test Hypothesis 3, on the role of contact profiles in loneliness trajectories. For boys, loneliness did not change for the *low contact* profile ($M_S = 0.02$, [-0.14; 0.18]). Loneliness decreased over time for boys in the *high all-round contact* profile ($M_S = -0.07$, [-0.10; -0.05]) and boys in the *high contact but low (video) calling* profile ($M_S = -0.08$, [-0.11; -0.05]). For girls, loneliness decreased for the *high all-round contact* profile ($M_S = -0.07$, [-0.10; -0.04]) and, quite unexpectedly, also for the *low contact* profile ($M_S = -0.26$, [-0.43; -0.09]). We did not find support for a change in loneliness for girls in the *high contact but low (video) calling* profile ($M_S = -0.11$, [-0.23; 0.01]). Taken together, even though no increase in loneliness was detected among students in low contact groups (as argued in Hypothesis 3), these groups were likely to remain stable in loneliness in a period during which most students decreased in loneliness. Comparing confidence intervals revealed no significant differences between the slopes of the three contact profiles for either boys or girls. Fig 3 shows the trajectories of loneliness for each contact profile for boys and girls separately.

Besides examining the degree of change using LGCA, we conducted additional (multivariate) analyses of covariance to examine how contact profiles differed in loneliness per time point. The findings revealed that all contact profiles differed in (combined time points of)

**Table 5. Overlap between peer status groups and contact profiles in percentages.**

| | Low contact (*n* = 31) | High contact but low (video) calling (*n* = 100) | High all-round contact (*n* = 374) |
|---|---|---|---|
| Normative | 67.7% | 86.0% | 84.0% |
| Rejected | 25.8% | 10.0% | 11.3% |
| Victimized | 6.5% | 4.0% | 4.8% |

*Note.* Columns add up to 100

**Table 6. Intercepts and slopes of the multi-group LGCA on loneliness per contact profile by gender (N = 505).**

| Gender | Parameter | Contact Profile | | |
|--------|-----------|-----------------|---|---|
| | | Low contact | High contact but low (video) calling | High all-round contact |
| Boys | $M$ intercept | 0.258 [0.078; 0.438]* | 0.159 [0.110; 0.207]*** | 0.215 [0.153; 0.278]*** |
| | $M$ slope | 0.018 [-0.141; 0.177] | -0.079 [-0.110; -0.048]*** | -0.072 [-0.095; -0.049]*** |
| Girls | $M$ intercept | 0.741 [0.399; 1.082] *** | 0.417 [0.178; 0.655] *** | 0.285 [0.226; 0.343] *** |
| | $M$ slope | -0.259 [-0.431; -0.086] ** | -0.109 [-0.229; 0.012] | -0.069 [-0.099; -0.040] *** |

*Note.* *$p < .05$

**$p \leq .01$

***$p \leq .001$. Numbers in parentheses represent 95% confidence intervals.

loneliness when gender was included ($F(6, 970) = 6.07$, $p < .001$, Wilks' = 0.93). Analysis of covariance showed that this difference existed at all time points: in Jan/Feb '20: $F(2, 496) = 4.83$, $p = .008$, $\eta p^2 = .02$, March-May '20 (measured retrospectively): $F(2, 500) = 14.74$, $p < .001$, $\eta p^2 = .06$, and in Oct/Nov '20: $F(2, 491) = 3.192$, $p = .042$, $\eta p^2 = .01$. Post-hoc tests using a Bonferroni adjustment showed that in Jan/Feb '20, students in the *low contact* profile reported higher levels of loneliness ($M = 0.46$) than students in the *high all-round contact* profile ($M = 0.25$, $p = .015$), whereas the *high contact but low (video) calling* profile ($M = 0.29$) did not differ significantly from the two other profiles. In March-May '20, the *low contact* profile reported the highest levels of loneliness ($M = 0.57$), compared with the *high contact but low (video) calling* profile ($M = 0.14$) and the *high all-round contact* profile ($M = 0.20$, both $p < .001$). We did not find differences in levels of loneliness in March-May '20 between the *high contact but low (video) calling* profile and the *high all-round contact* profile ($p = 1.00$). Loneliness in Oct/Nov '20 showed a similar pattern to Jan/Feb '20, with students in the *low contact* profile reporting higher levels of loneliness ($M = 0.24$) than students in the *high all-round contact* profile ($M = 0.11$, $p = .036$). Students in the *high contact but low (video) calling* profile ($M = 0.09$) did not report different levels of loneliness from students in the *low* ($p = .091$) and *high all-round contact* profiles ($p = 1.00$). Students who had low contact with friends during the lockdown thus reported higher levels of loneliness.

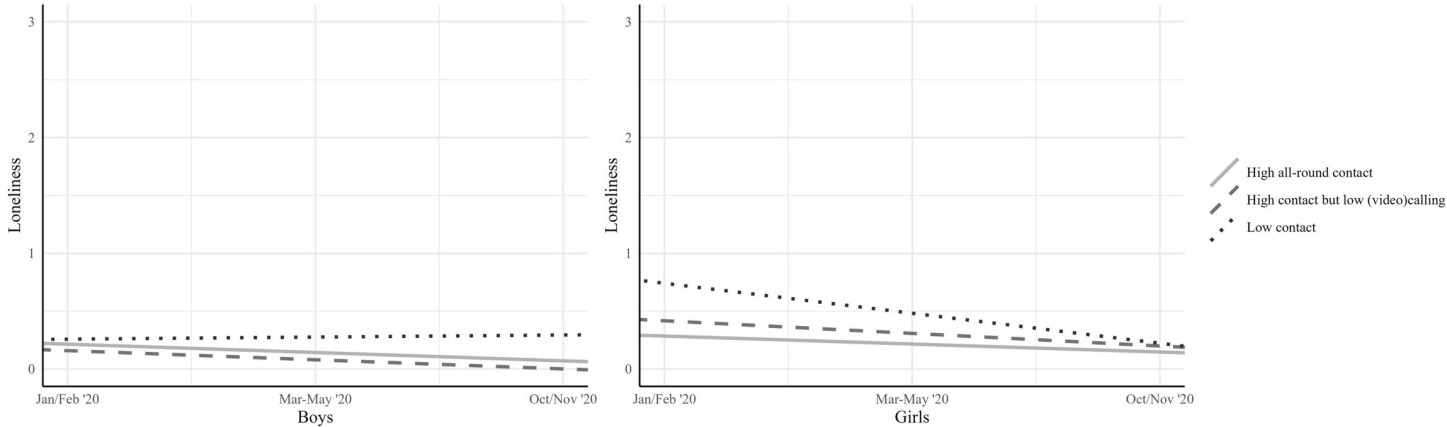

**Fig 3.** Trajectories of loneliness per contact profile for boys (left) and girls (right). *Note.* Loneliness during the first lockdown in March-May 2020 was measured retrospectively in October/November 2020.

## Discussion

In the current study, we (1) examined how peer-related loneliness developed among young adolescents during the first lockdown of the COVID-19 pandemic and (2) identified who was most at risk for loneliness in terms of peer status and social contact. Unexpectedly, the results showed that loneliness declined for the students in our sample during the first lockdown in the Netherlands (March-May 2020). Although the average levels of loneliness decreased, this was not the case for students who had low contact with friends during the school shutdown, revealing this group to be the most at risk for loneliness. Students in the victimized and rejected groups had the largest decline in loneliness, suggesting that these students experienced temporary relief during the lockdown from the negative peer experiences that usually contributed to their feelings of loneliness.

### Average development of loneliness

The adolescents in this study did not experience an increase in loneliness during the first lockdown of the COVID-19 pandemic, but rather a decrease; this was contrary to Hypothesis 1. Three explanations for this decrease are likely, which may be tested in future research. First, following studies on loneliness among adults, the pandemic may have evoked the feeling that 'we are all in the same boat' [17]. The restricted opportunities for physical social contact may have led to lower expectations of social contact, and at the same time may have resulted in elevated feelings of solidarity and connectedness, which in turn may have led to decreased feelings of loneliness. Thus, loneliness may have decreased during the pandemic because it felt less lonely being lonely together.

Second, the pandemic may have presented opportunities to be selective in one's contacts, which may have led to a decrease in the experience of negative social contacts. In line with this reasoning, young adolescents in a similar sample did not experience an increase in stress, depression, or anxiety during the first lockdown [9, 25]. The majority of students in our sample kept daily to weekly in-person contact with friends during the first lockdown, but probably had fewer negative peer experiences.

Third, the decrease in loneliness may be specific to students in the developmental period that was the focus of our study. That is, our participants transitioned from primary to secondary school, which gave them the opportunity to improve their peer relationships. Yet, most students have fewer friends, are more socially isolated, and become more lonely after transitioning to secondary school [21, 22]. Furthermore, the decrease in loneliness from the period before to the period during the school shutdown cannot be explained by school transition effects as the students were still in primary education at the time of the lockdown (March-May '20). Thus, as the decrease in loneliness cannot solely be attributed to the school transition, it speaks to reason that there was an effect of the pandemic. Moreover, the pandemic may have increased loneliness for older adolescents more than for early adolescents. For instance, undergraduate students in Switzerland felt more lonely during the pandemic than before the pandemic [45], and loneliness increased as the pandemic progressed for German undergraduate students [46]. Students more often lived by themselves with less comfortable living conditions. Feeling confined to their room, they lacked a sense of belonging in their new neighborhood and institution, and they were hindered from navigating life transitions, such as forming romantic relationships [47]. Early adolescents, including those in our sample, possibly had more access to structural social contact during the lockdown, including living with parents and siblings, and seeing neighborhood friends. As the pandemic progressed, physical distancing behaviors and loneliness increased [48]. Future studies should assess possible rebound effects, chronic loneliness, and the long-term negative effects of the pandemic for adolescents.

## The role of peer status

This study examined who is most at risk for loneliness in terms of peer status and social contact, because peer-related factors may partially predict heterogeneity in the impact of the pandemic on loneliness. We found *normative*, *rejected*, and *victimized* peer status groups, reflecting students' in-class peer status prior to the pandemic. Whereas previous studies identified a *popular-liked* group [23, 49], this group was not detected in our sample, for two possible reasons. First, the sample might have been too small to detect a *popular-liked* group, as LCA generally detects more classes in larger samples. Second, most previous studies measured peer status in secondary education, whereas this study focused on peer status in primary education. A popular-liked group is possibly only found in secondary education, because peer status becomes more complicated and crystallized in secondary education. In that case, being liked is normative in primary education, and the main exceptions to the normative group are students with a negative peer status.

Trajectories of loneliness were specified per peer status group. Students in the *victimized* group had the highest initial levels of loneliness, followed by the *rejected* and the *normative* groups. When trajectories per peer status group were specified, the development of loneliness did not differ much. Loneliness declined for all peer status groups. Students with a higher peer status were clustered in the *normative* group. These students also decreased in their feelings of loneliness, contrary to Hypothesis 2a and 2b. This may be for the same reasons as the general decline in loneliness, as discussed above. When trajectories of loneliness were explored for students with a lower peer status, it was found that, for boys, the decline in loneliness was larger for the *victimized* peer status group than for the *rejected* and the *normative* groups. For girls, the decline in loneliness was larger for the *rejected* and the *victimized* groups than for the *normative* group. Thus, in general, loneliness was most likely to decrease among students in low status groups. Students with higher initial levels of loneliness possibly tend to have more rapid decreases in loneliness. This decline might also suggest that students who were victimized or rejected prior to the pandemic benefited from the lockdown. Nevertheless, we found that students in the *victimized* group were generally lonelier than those in the *normative* group at every time point. Thus, despite the absolute within-group decline of loneliness during the first lockdown for victimized students, these students remained at risk for loneliness compared with students with a normative peer status. Future studies could oversample victimized and rejected students to confirm these effects in larger samples.

## The role of contact with friends

Exploring profiles of type (i.e., offline and online) and amount of social contact with friends during the school shutdown yielded a *low contact*, a *high contact but low (video) calling*, and a *high all-round contact* profile. Whereas loneliness decreased over time for students in the *high all-round contact* profile, it did not decrease for boys in the *low contact* profile, nor for girls in the *high contact but low (video) calling* profile; this is largely in line with Hypothesis 3. Moreover, students in the *low contact* profile and the *high contact but low (video) calling* profile reported the highest levels of loneliness at all time points. Thus, students who had low contact with friends during the first lockdown were the most at risk for loneliness. This is in line with the findings of previous studies, which revealed that students who spent less time with peers during the pandemic felt more lonely [27, 28]. At the same time, although connecting with friends online combatted loneliness, it did not increase adolescents' feelings of happiness [50], and it was related to greater depression [7]. Therefore, possibilities for in-person contact with friends during the pandemic may have been the most promising to combat loneliness.

## Strengths and limitations

This study has several strengths compared with previous research. We used a longitudinal design including a baseline measure before the pandemic, examined at-risk groups for loneliness beyond average levels of loneliness, and used a scale for peer-related loneliness rather than a single-item measure. In addition, we examined a possibly vulnerable group of early adolescents transitioning from primary to secondary education, and used a relatively representative sample rather than relying on responses to online surveys. Despite this, the results must be interpreted with the following limitations in mind. First, a retrospective measure was used to measure loneliness for March-May '20, during the first lockdown. This may have led to a different interpretation of the loneliness measure at this time point, as suggested by the unacceptable fit of the scalar step of the measurement invariance test. The students may have had biased perceptions and evaluated their loneliness during the lockdown as disproportionally low, as they may have put the severity of their loneliness during the lockdown in perspective after the relaxation of the COVID-19 measures. On the other hand, the retrospective measure may have caused a benchmark effect, because the students may have valued their peer relationships at school more after missing them for a while. This may have led to disproportionally high reported levels of loneliness during the lockdown and school shutdown (March-May '20), and low levels after the reopening of school at the time of measurement (Oct/Nov '20). As a result, the levels of loneliness for March-May '20 should be interpreted with caution. Despite the use of the retrospective measure, the risk that the definition of loneliness would change over time was limited by our use of a scale containing specific items (i.e., 'I feel sad that I have no friends'), rather than relying on a more subjective single-item measure for loneliness. That the same construct was measured over time was confirmed by the acceptable fit of the configural step of the measurement invariance test. Moreover, our longitudinal study, consisting of three time points of which one was retrospective, gives more information than a cross-sectional design and, therefore, builds upon the existing literature on the COVID-19 pandemic.

Second, peer status and contact with friends were measured at only one time point. Peer status was only measured in Jan/Feb '20, in primary education before the COVID-19 pandemic. Yet, the students in the PRIMS study transitioned from primary to secondary education, where they entered a new peer group, which may have changed their peer status, and in turn changed their feelings of loneliness. For instance, students who were victimized in primary education may have improved their social status and established new friendships in secondary education, decreasing their feelings of loneliness. Insight into changes in peer status would clarify the co-evolution of peer status and loneliness [23]. Future studies would add to the literature by examining the (dis)continuation of peer experiences over the transition from primary to secondary education. We had information on students' contact with friends only for March-May '20, during the first lockdown. Therefore, we were unable to compare students' contact with friends during the pandemic with their contact before the pandemic. Future studies could keep track of students' contact with friends as the pandemic ebbed, to examine changes in contact as physical contact became less restricted. In addition, future studies could investigate the nature and quality of contact with friends: for instance, co-rumination about the negative impact of the pandemic may have had an adverse effect on students' well-being.

## Practical implications

The findings of this study have implications for policy, school practice, and parents for during and after the COVID-19 pandemic. On average, loneliness declined during and directly after the first lockdown (March-May 2020) in our sample of young adolescents. Despite this average decline in loneliness, some youth reported higher levels of loneliness during the first lockdown

than others. Students who had a low level of contact with friends during the school shutdown felt the most lonely. These students may have benefited the most from interventions combatting loneliness, not only *during* the pandemic but also while *recovering from* the pandemic. In similar future circumstances, parents can encourage their children to maintain contact with friends during the pandemic to combat loneliness. Where possible, children can use opportunities for in-person contact with friends, such as by physically distancing and meeting outside. In case of a severe lockdown, parents can make it possible for their children to maintain online contact with friends, such as by chat. Policy makers should be aware of the adverse effects of school shutdowns not only for students' academic development [51], but also for their socio-emotional well-being. Policy makers might modify physical isolation measures for adolescents, as the impact of such measures might be particularly negative for this age group. For instance, policy makers might prioritize opening schools or relax physical isolation measures for adolescents before other age groups.

Students who were victimized or rejected before the pandemic experienced the largest decrease in loneliness over the first lockdown. These students may have experienced a 'break' from the negative peer interactions at school that usually contributed to their feelings of loneliness. Likewise, victimization in schools substantially decreased during the school lockdown, with cyberbullying not increasing [52]. Students who were victimized prior to the pandemic experienced more teacher support and liked remote schooling more than other students did [52]. Victimization was less prevalent among students who spent more time in online schooling during the pandemic compared with offline schooling, and victimization-related anxiety was lower when students attended online schooling [53]. Moreover, on average, the students in our sample felt less excluded, abandoned, and left out by peers during the school shutdown. This suggests that having fewer negative peer experiences was a positive side-effect of the lockdown, which points to the challenges for adolescents of functioning within a classroom. As schools reopen following such a shutdown, school practitioners should be aware of the social challenges for rejected and victimized students, because these students may have experienced online schooling as socially safe. As the pandemic ebbs, interventions should focus on preventing students from being victimized and rejected in order to combat loneliness. For instance, evidence-based anti-bullying programs are promising. Furthermore, interventions specifically focused on combatting loneliness should be further developed [54].

## Supporting information

**S1 Appendix. Latent class analysis and multi-group latent growth curve analysis peer status groups.**
(DOCX)

**S2 Appendix. Latent class analysis and multi-group latent growth curve analysis contact profiles.**
(DOCX)

## Author Contributions

**Conceptualization:** Sofie J. Lorijn, Maaike C. Engels, Gerine M. A. Lodder, René Veenstra.

**Data curation:** Sofie J. Lorijn.

**Formal analysis:** Sofie J. Lorijn, Lydia Laninga-Wijnen.

**Funding acquisition:** René Veenstra.

**Investigation:** Sofie J. Lorijn.

**Methodology:** Sofie J. Lorijn, Lydia Laninga-Wijnen, Maaike C. Engels.

**Project administration:** Sofie J. Lorijn.

**Software:** Sofie J. Lorijn, Lydia Laninga-Wijnen.

**Supervision:** Lydia Laninga-Wijnen, René Veenstra.

**Visualization:** Sofie J. Lorijn.

**Writing – original draft:** Sofie J. Lorijn.

**Writing – review & editing:** Sofie J. Lorijn, Lydia Laninga-Wijnen, Gerine M. A. Lodder, René Veenstra.

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
