## [Decision Letter · Decision Letter 0]

8 Mar 2023

PONE-D-23-02074The Development of Adolescents’ Loneliness during the COVID-19 Pandemic:

The Role of Peer Status and Contacts with FriendsPLOS ONE

Dear Dr. Lorijn,

Thank you for submitting your manuscript to PLOS ONE. After careful consideration, we feel that it has merit but does not fully meet PLOS ONE’s publication criteria as it currently stands. Therefore, we invite you to submit a revised version of the manuscript that addresses the points raised during the review process.

We look forward to receiving your revised manuscript.

Kind regards,

Giulia Ballarotto

Academic Editor

PLOS ONE

Journal Requirements:

4. Please ensure that you include a title page within your main document. You should list all authors and all affiliations as per our author instructions and clearly indicate the corresponding author.

5. Please amend your manuscript to include your abstract after the title page.

Additional Editor Comments:

As you can see from the reviewers' comments, some changes are suggested, which I agree with. In addition, I stress the importance of reviewing the text with a native English speaker.

Reviewers' comments:

Reviewer's Responses to Questions

**Comments to the Author**

1. Is the manuscript technically sound, and do the data support the conclusions?

Reviewer #1: Yes

Reviewer #2: Yes

2. Has the statistical analysis been performed appropriately and rigorously? 

Reviewer #1: Yes

Reviewer #2: Yes

3. Have the authors made all data underlying the findings in their manuscript fully available?

Reviewer #1: No

Reviewer #2: No

4. Is the manuscript presented in an intelligible fashion and written in standard English?

Reviewer #1: Yes

Reviewer #2: Yes

5. Review Comments to the Author

Reviewer #1: I read with great interest the Manuscript titled “The Development of Adolescents’ Loneliness during the COVID-19 Pandemic: The Role of Peer Status and Contacts with Friends” (PONE-D-23-02074), which falls within the aims of PLOS ONE.

In my honest opinion, the study is well written and the topic is interesting enough to attract the readers’ attention. However, the authors should improve the discussion by citing other relevant articles about the topic.

Authors should consider the following recommendations:

- Does this online survey conform to the recommended standards for conducting and reporting web-based surveys, i.e., the Checklist for Reporting Results of Internet E-surveys (CHERRIES)? I suggest specifying this point.

- I recommend further improving the references about the impact of the COVID-19 pandemic on adolescents worldwide by citing some of these recent studies on the topic: DOI: 10.3389/fpsyg.2020.559951; DOI: 10.1016/j.jad.2021.04.016; DOI: 10.1089/cyber.2020.0478.

- It would be interesting to compare the experience of adolescents with that of university students to highlight similarities and differences. In this regard, it may be helpful for authors to read some recently published articles on this topic: DOI: 10.1016/j.psychres.2020.113111; DOI: 10.13129/2282-1619/mjcp-3009; DOI: 10.1038/s41598-022-21288-z.

Reviewer #2: I have read with great interest the manuscript "The Development of Adolescents’ Loneliness during the COVID-19 Pandemic:

The Role of Peer Status and Contacts with Friends" which addresses a very important topic linked to the impact that COVID-19 had on adolescents.

I really enjoyed reading the manuscript, however, I believe that there are several shortcomings. I will list them below in no particular order of importance.

1. Please revise the English. Even if the manuscript is overall easy to follow and understand, there are some areas where the authors could polish the phrasing. For example, at line 6 to 7 "These societal concerns were of special concern during the COVID-19 pandemic" I would encourage the authors to find a synonym for "concern", perhaps. This is just an example, please revise the whole manuscript. Furthermore, I would encourage the authors to revise the introduction section. I would welcome a somewhat more robust argumentation of the importance of this study. Also, it could be helpful if the overall introduction was revised - the arguments provided by the authors seem somewhat fragmented, I would want to see more coherence and flow.

2. If you choose to provide graphics, please make sure that they are easy to read. The ones provided are very blurry and hard to follow. Also, I would replace figure 3 and 4 with tables containing percentages/means. This is somewhat a personal preference, as I find more easy to follow a table than a graph. If the authors choose to retain the graphs, please add the percentages/means to each graph.

3. I would welcome a more robust introduction with a clearer emphasis on the importance of the study. I would also try to be more concise, as at times I had the impression of reading sections of the procedure/measurement.

4. The discussion section conveys the personal interpretation of the authors, however, I would like to see a more theoretically based interpretation of the findings. Furthermore, I would welcome further practical implications of the findings.

5. In the Results - Descriptive statistics section, there are some things I would welcome more details about. For example, the authors mention "Stability in students’ feelings of loneliness was moderate to strong". What does this mean? I see a correlation afterwards and I imagine that it is the correlation for LACA scores from T1 correlated to T2 and T2 with T3 (measures)? Also I am puzzled by the choice of words, stability. It is highly expected that the scores from the same instrument correlate.. Maybe there is a nuance there that the authors could shed some light on.

6. In the Results - Descriptive statistics section, the authors say "Although measured in the same survey, feelings of loneliness differed between March-May ‘20 and Oct/Nov ‘20", and then provide a correlation score. In the light of the previous understanding of the statistics linked to the stability, I have to admit that this is confusing. Furthermore, when differences are mentioned, I usually expect a T test, not necessarily a correlation. Could the authors indicate the reasoning/rephrase this section?

6. PLOS authors have the option to publish the peer review history of their article (what does this mean?). If published, this will include your full peer review and any attached files.

Reviewer #1: No

Reviewer #2: No

---

## [Author Response · Author response to Decision Letter 0]

4 Apr 2023

We are grateful for the constructive feedback and we considered and in almost all cases followed reviewers recommendations. For the response to your comments, please see the documents 'response to reviewers' and 'revised manuscript'.

---

## [Decision Letter · Decision Letter 1]

9 May 2023

The Development of Adolescents’ Loneliness during the COVID-19 Pandemic:

The Role of Peer Status and Contact with Friends

PONE-D-23-02074R1

Dear Dr. Lorijn,

We’re pleased to inform you that your manuscript has been judged scientifically suitable for publication and will be formally accepted for publication once it meets all outstanding technical requirements.

Kind regards,

Giulia Ballarotto

Academic Editor

PLOS ONE

Additional Editor Comments (optional):

Reviewers' comments:

Reviewer's Responses to Questions

**Comments to the Author**

1. If the authors have adequately addressed your comments raised in a previous round of review and you feel that this manuscript is now acceptable for publication, you may indicate that here to bypass the “Comments to the Author” section, enter your conflict of interest statement in the “Confidential to Editor” section, and submit your "Accept" recommendation.

Reviewer #1: All comments have been addressed

Reviewer #2: All comments have been addressed

2. Is the manuscript technically sound, and do the data support the conclusions?

Reviewer #1: (No Response)

Reviewer #2: Yes

3. Has the statistical analysis been performed appropriately and rigorously? 

Reviewer #1: (No Response)

Reviewer #2: Yes

4. Have the authors made all data underlying the findings in their manuscript fully available?

Reviewer #1: (No Response)

Reviewer #2: No

5. Is the manuscript presented in an intelligible fashion and written in standard English?

Reviewer #1: (No Response)

Reviewer #2: Yes

6. Review Comments to the Author

Reviewer #1: (No Response)

Reviewer #2: (No Response)

7. PLOS authors have the option to publish the peer review history of their article (what does this mean?). If published, this will include your full peer review and any attached files.

Reviewer #1: No

Reviewer #2: No

---

## [Editor Report · Acceptance letter]

19 May 2023

PONE-D-23-02074R1 

The development of adolescents’ loneliness during the COVID-19 pandemic:
The role of peer status and contact with friends 

Dear Dr. Lorijn:

I'm pleased to inform you that your manuscript has been deemed suitable for publication in PLOS ONE. Congratulations! Your manuscript is now with our production department. 

Kind regards, 

on behalf of

Dr Giulia Ballarotto 

Academic Editor

PLOS ONE